# Allosteric Interactions between Adenosine A_2A_ and Dopamine D_2_ Receptors in Heteromeric Complexes: Biochemical and Pharmacological Characteristics, and Opportunities for PET Imaging

**DOI:** 10.3390/ijms22041719

**Published:** 2021-02-09

**Authors:** Kavya Prasad, Erik F. J. de Vries, Philip H. Elsinga, Rudi A. J. O. Dierckx, Aren van Waarde

**Affiliations:** 1Department of Nuclear Medicine and Molecular Imaging, University Medical Center Groningen, University of Groningen, Hanzeplein 1, 9713GZ Groningen, The Netherlands; e.f.j.de.vries@umcg.nl (E.F.J.d.V.); p.h.elsinga@umcg.nl (P.H.E.); r.a.dierckx@umcg.nl (R.A.J.O.D.); 2Department of Diagnostic Sciences, Ghent University Faculty of Medicine and Health Sciences, C.Heymanslaan 10, 9000 Gent, Belgium

**Keywords:** adenosine A_2A_ receptor, dopamine D_2_ receptor, heteromers, allosteric interaction, receptor–receptor interactions, striatum, GABAergic enkephalinergic neuron

## Abstract

Adenosine and dopamine interact antagonistically in living mammals. These interactions are mediated via adenosine A_2A_ and dopamine D_2_ receptors (R). Stimulation of A_2A_R inhibits and blockade of A_2A_R enhances D_2_R-mediated locomotor activation and goal-directed behavior in rodents. In striatal membrane preparations, adenosine decreases both the affinity and the signal transduction of D_2_R via its interaction with A_2A_R. Reciprocal A_2A_R/D_2_R interactions occur mainly in striatopallidal GABAergic medium spiny neurons (MSNs) of the indirect pathway that are involved in motor control, and in striatal astrocytes. In the nucleus accumbens, they also take place in MSNs involved in reward-related behavior. A_2A_R and D_2_R co-aggregate, co-internalize, and co-desensitize. They are at very close distance in biomembranes and form heteromers. Antagonistic interactions between adenosine and dopamine are (at least partially) caused by allosteric receptor–receptor interactions within A_2A_R/D_2_R heteromeric complexes. Such interactions may be exploited in novel strategies for the treatment of Parkinson’s disease, schizophrenia, substance abuse, and perhaps also attention deficit-hyperactivity disorder. Little is known about shifting A_2A_R/D_2_R heteromer/homodimer equilibria in the brain. Positron emission tomography with suitable ligands may provide in vivo information about receptor crosstalk in the living organism. Some experimental approaches, and strategies for the design of novel imaging agents (e.g., heterobivalent ligands) are proposed in this review.

## 1. Introduction

Adenosine, a purine nucleoside, plays several behavioral and physiological roles throughout the central nervous system (CNS). Adenosine is generated in the living brain from adenine nucleotides such as adenosine triphosphate (ATP) and adenosine monophosphate (AMP). A much less important, other source of adenosine is S-adenosylhomocysteine, that originates from S-adenosylmethionine after physiological transmethylation [1]. Increased firing of neurons is associated with increased consumption of ATP, nucleotide dephosphorylation, and increases of intracellular adenosine levels (Figure 1). Since equilibrative nucleoside transporters are present in neuronal membranes, the extracellular levels of adenosine will also increase under such conditions. Thus, extracellular adenosine concentrations fluctuate, depending on neuronal activity.

Extracellular adenosine levels in the mammalian brain range from 20 to 250 nM [2,3,4,5,6,7]. Extracellular adenosine can bind to four subtypes of adenosine receptors, called A_1_, A_2A_, A_2B_ and A_3_, which belong to the P1 receptor family. A_1_ and A_2A_ receptors have a high affinity for adenosine (10–100 nM range), whereas A_2B_ and A_3_ receptors are only activated when extracellular adenosine reach very high (micromolar) levels, after tissue damage (e.g., inflammation, hypoxia, ischemia, brain injury). Physiological levels of adenosine will stimulate the A_1_ and A_2A_ receptors. It is unlikely that adenosine exerts major physiological functions via A_2B_ and A_3_ receptors in the brain, since physiological levels of adenosine are too low to activate these proteins, and A_2B_ and A_3_ receptors are mainly expressed in peripheral organs rather than in the CNS [8,9,10,11,12,13].

A_1_ receptors (A_1_R) are coupled to Gi proteins. Stimulation of these receptors by adenosine causes a decrease in cAMP levels through an inhibitory effect on adenylate cyclase. A_2A_ receptors (A_2A_R) are coupled to an excitatory Gs protein. Stimulation of A_2A_R results in an increase of cAMP levels and activation of protein kinase A [8,9,10,11,12,13].

## 2. Antagonistic Interactions between Adenosine and Dopamine

### 2.1. Living Animals

Interactions between adenosine and dopamine in living animals were already observed in 1974. Adenosine antagonists (caffeine and theophyllamine) were then reported to enhance the action of dopamine agonists such as apomorphine, bromocriptine and L-DOPA (stimulation of rotation behavior) in the 6-hydroxydopamine hemiparkinson model of rats [14]. In later studies using reserpinized (i.e., dopamine-depleted) mice, the action of bromocriptine was found to be inhibited by adenosine agonists (L-PIA, NECA) and this inhibition could be reversed by the adenosine antagonists caffeine, paraxanthine, and theophylline. Since the non-subtype-selective agonist 5’-(N-ethyl)carboxamido-adenosine (NECA) was considerably more potent than the A_1_-selective agonist N6-R-phenylisopropyladenosine (L-PIA), A_2A_ rather than A_1_ receptors seem to be involved in the inhibition of the locomotor response to dopaminergic stimulation [15,16]. Central administration of the adenosine A_2A_R agonist 2-[p-(2-carboxyethyl)phenethylamino]-5’-N-ethylcarboxamido-adenosine (CGS21680) was shown to induce catalepsy in the rat, and this effect was counteracted by systemic administration of the adenosine antagonist theophylline or the dopamine D_2_ agonist 5,6,7,8-Tetrahydro-6-(2-propen-1-yl)-4H-thiazolo[4,5-d]azepin-2-amine dihydrochloride (BHT-920) [17]. The dopamine D_2_R antagonist haloperidol induces catalepsy and Parkinsonian symptoms in rats and mice. Such symptoms can be reversed by treating rats with the non-selective adenosine antagonist caffeine or the selective A_2A_R antagonist SCH58261 [18] and are significantly reduced in A_2A_R knockout mice [19]. Haloperidol-induced motor impairments in monkeys (catalepsy, extrapyramidal syndrome) are counteracted by the A_2A_R antagonists SCH-412348, istradefylline, and caffeine [20]. 

As A_2A_ receptors are known to be located mainly in the striatum, in postsynaptic locations on dendrites and dendritic spines [21,22] and, to a lesser extent (25%), on nerve endings [23,24]. These findings suggest the existence of postsynaptic interactions between adenosine and dopamine receptors, probably the A_2A_ and D_2_ subtypes. Stimulation of A_2A_ receptors results in inhibition, and blockade of A_2A_ receptors in enhancement of D_2_-receptor mediated locomotor activation. 

Stimulation of A_2A_R in the nucleus accumbens of rats by local infusion of CGS21680 produced behavioral effects similar to those induced by local dopamine depletion (i.e., decreased lever pressing for preferred food and substantially increased consumption of the less preferred but freely accessible chow) [25]. On the other hand, decreases of lever pressing for preferred (high carbohydrate) food caused by the D_2_R antagonist eticlopride could be partially reversed by treating rats with the A_2A_R antagonist MSX-3 [26]. Similar decreases induced by the D_2_R antagonist haloperidol could be reversed by the A_2A_R-subtype-selective antagonist istradefylline or the non-subtype selective AR antagonist caffeine [27]. Thus, antagonistic interactions between A_2A_R and D_2_R occur not only in the dorsal striatum where they control locomotor activity, but also in the nucleus accumbens (ventral striatum) where they affect goal-directed behavior.

### 2.2. Membrane Preparations

Antagonistic interactions between A_2A_ and D_2_ receptors could also be observed in vitro, in membrane preparations from rat striatum. Administration of the adenosine A_2A_ receptor (A_2A_R) agonist CGS21680 resulted in a significant, 40% increase of the K_d_ (i.e., a loss of the affinity) of dopamine D_2_ receptors to the agonist L-(-)-N-[^3^H]propylnorapomorphine without changing the B_max_ (i.e., the number of D_2_ receptors) [28]. However, the K_d_ and B_max_ for binding of the dopamine D_2_ antagonist [^3^H]raclopride were not affected [28]. The effect of CGS21680 on D_2_R affinity was most pronounced at concentrations similar to the K_d_ for binding of CGS21680 to A_2A_R. At very high, saturating doses of CGS21680 (300 nM), the effect of the agonist was reduced, probably because such high doses cause a desensitization of A_2A_R [28]. In striatal membrane preparations of adult (as opposed to young) rats, CGS21680 reduced not only the affinity of D_2_ receptors for agonists, but also the fraction of D_2_ receptors in the high-affinity state. Thus, A_2A_R stimulation may inhibit the motor responses induced by dopamine receptor agonists by decreasing both the affinity and the signal transduction of D_2_ receptors [29,30]. Adenosine appears to regulate the properties of D_2_R via its interaction with A_2A_R. Direct receptor–receptor interactions in striatal membranes were suggested as a potential mechanism involved in this pharmacological crosstalk between A_2A_R and D_2_R [28,31,32].

### 2.3. Intact Cells

Antagonistic interactions between A_2A_ and D_2_ receptors were also demonstrated in intact cells. In a mouse fibroblast cell line stably transfected with A_2A_R and D_2_R, the D_2_R agonist quinpirole induced a concentration-dependent increase in intracellular (cytosolic) free calcium. This response was completely blocked if cells were pretreated with haloperidol. CGS21680 by itself did not affect intracellular calcium levels (even when it was administered at high dose), but CGS21680 strongly counteracted the response of [Ca^2+^]_i_ to quinpirole [33]. Similar observations were made in SH-SY5Y (human neuroblastoma) cells that were transfected with human D_2_R [34]. The effect of CGS21680 was shown to be related to a two- to three-fold decrease of the affinity of the D_2_R in the cells to dopamine receptor agonists [34,35]. A similar three- to four-fold increase of the K_D_ of dopamine at high-affinity D_2_R sites after administration of CGS21680 was noted in Chinese hamster ovary (CHO) cells that were co-transfected with A_2A_ and D_2_ receptors [36]. In such cells, CGS21680 decreased the affinity of D_2_ receptors for [^3^H]dopamine but not the number of dopamine binding sites [37]. Since A_2A_R stimulation increases, but D_2_R stimulation decreases, the intracellular formation of cyclic AMP, A_2A_R, and D_2_R may interact not only at the membrane level but also at the second messenger level. The experiments in CHO cells suggested that the latter interaction may be quantitatively the most important [36]. 

In initial cell experiments, A_2A_R agonists were shown to decrease the affinity of D_2_R for agonists. In later experiments, interactions in the opposite direction were also demonstrated. D_2_R activation by quinpirole resulted in a less rapid and reduced binding of the fluorescent A_2A_R agonist MRS5424 to HEK293 cells, which expressed both A_2A_ and D_2_ receptors [38]. Similar decreases of A_2A_R agonist binding were observed when the cells were treated with D_2_R agonists in clinical use, such as pramipexole, rotigotine, and apomorphine [39]. On the other hand, chronic D_2_R blockade by haloperidol increased both the affinity and the responsiveness of the A_2A_R to the agonist NECA in CHO cells that expressed both A_2A_ and D_2_ receptors [40]. 

In CHO cells transiently transfected with A_2A_R and D_2_R, both the A_2A_R agonist CGS21680 and the AR antagonist caffeine caused a decrease of the affinity of the D_2_R for radioligands, not only the D_2_R agonist [^3^H]quinpirole but also the D_2_R antagonist [^3^H]raclopride. Yet, CGS21680 and caffeine canceled out each other’s effect on D_2_R affinity when they were administered together [41]. These apparently paradoxical findings led to a novel hypothesis concerning the structural basis of adenosine–dopamine receptor interactions, which is described in Section 5 of this review.

### 2.4. Brain Slices

Antagonistic interactions between A_2A_ and D_2_ receptors could also be demonstrated in cryostat sections of rat and human brain. CGS21680 significantly increased the IC_50_ values of competition between the D_2/3_R ligand [^125^I]iodosulpiride and dopamine in the striatal region of such preparations [42].

## 3. Regional, Cellular, and Subcellular Distribution of A_2A_ and D_2_ Receptors

The antagonistic interactions of A_2A_ and D_2_ receptors that were observed in rat striatal membranes [28,29,30] suggested that the A_2A_ and D_2_ receptor genes are co-expressed by some cells in the mammalian brain.

### 3.1. Regional Distribution

Both in the rodent and human brain, A_2A_R mRNA [43,44,45,46,47,48] and A_2A_R protein [24,49,50,51,52,53,54,55,56] are mainly located in the striatum (caudate-putamen) and nucleus accumbens. In monkeys, A_2A_R immunoreactivity is mainly present in striatum and nucleus accumbens, but can also be detected in the substantia nigra, an area showing very low A_2A_R density in rats. This finding indicates that there may be species differences between rodents and primates concerning the regional distribution of A_2A_R [57]. 

Caudate, putamen and nucleus accumbens express also high numbers of dopamine D_2_R [58,59,60,61,62]. The distribution of D_2_R in the rodent brain is very similar to that of A_2A_R mRNA, although D_2_R is also present in the substantia nigra and piriform cortex [63]. 

### 3.2. Cellular Distribution

The majority (more than 95%) of the neurons in the striatum are medium spiny neurons (MSNs; i.e., medium-sized neurons (diameter 12–15 µm in rodents) with large and extensive dendritic trees) [64]. MSNs in the dorsal striatum can be divided in two subtypes [65,66]. Both subtypes use gamma-aminobutyric acid (GABA) as neurotransmitter, but the subtypes have different projection patterns and they express different receptors and neuropeptides. Some MSNs send direct (monosynaptic) projections to the substantia nigra and the globus pallidus internus. Based on this projection pattern, this subtype is said to form part of the “direct pathway” (Figure 2). MSNs of the direct pathway express dopamine D1R and the peptide dynorphin (together with substance P). Other MSNs are indirectly linked to the substantia nigra and the globus pallidus internus, via the globus pallidus externus and the subthalamic nucleus. Because of this distinctive projection pattern, they are said to form part of the “indirect pathway” (Figure 2). MSNs of the indirect pathway express dopamine D_2_R and the peptide enkephalin [63,67,68] (reviewed in [69]). 

In early publications, the A_2A_R (at that time still called RDC8) was shown to be present in medium-sized but not in large neurons of the dog and rat striatum [70]. In contrast to A_2A_R, dopamine D_2_R mRNA is present both in medium-sized and large neurons [71]. Later studies employed double in situ hybridization [43,44,47,72,73] and double-labeling immunohistochemistry [52] to determine the phenotype of A_2A_R-containing neurons in dorsal and ventral striatum. In the ventral striatum, a population of neurons expresses the gene for the A_2A_R, but not for preproenkephalin. This sub-population is absent in the dorsal striatum. In the dorsal striatum, 95–96% of the A_2A_R mRNA is co-expressed with D_2_R mRNA. Only a few neurons expressing 3–6% of the A_2A_R mRNA co-express dopamine D_1_R or substance P mRNAs. In the ventral striatum, most A_2A_R mRNA (89–92%) co-localizes with preproenkephalin A mRNA, and the vast majority (93–95%) with D_2_R mRNA [74]. Adenosine A_2_A receptors were shown to co-localize with enkephalin and dopamine D_2_R, but not with dopamine D_1_R, substance P or somatostatin. These data were interpreted as evidence for a preferential expression of A_2A_R in striatopallidal GABAergic MSNs of the indirect pathway, cells which also express D_2_R [44,72,75]. Microdialysis experiments in intact freely moving rats supported this hypothesis. In these experiments, adenosine and dopamine agonists and antagonists were infused in the striatum, either alone or in combination, and the effect on the release of GABA was measured in the ipsilateral globus pallidus [76].

MSNs from the indirect pathway are the main, but not the only, cells in the striatum that co-express A_2A_ and D_2_ receptors. Striatal astrocytes also express both proteins [77,78,79,80] and receptor–receptor interactions between A_2A_R and D_2_R have been demonstrated in glia. Administration of the D_2_R agonist quinpirole to rat striatal astrocytes inhibits the 4-aminopyridine-provoked release of glutamate. The A_2A_R agonist CGS21680 alone did not affect glutamate release but reduced the D_2_R-mediated inhibiting effect of quinpirole [81]. A third class of cells in the striatum which express both A_2A_R and D_2_R are cholinergic interneurons [82].

### 3.3. Subcellular Location

In bright field photomicrographs of coronal sections of rat striatum, A_2A_R protein was detected on the cell bodies of GABA/enkephalin striatopallidal neurons [73]. Using immuno-electron microscopy, A_2A_Rs were mainly detected on dendrites, to a lesser extent on axon terminals, soma and astrocytic processes [23,24,52]. Subcellular fractionation experiments using the radioligand [^3^H]SCH58261 suggested that A_2A_R in the striatum of the rat are not enriched in synaptosomes [22]. In dendrites and soma, A_2A_R were shown to be present not only on the plasmalemma, but also throughout the cytoplasm and around intracellular membranous structures [23]. The predominantly postsynaptic location of A_2A_Rs (on dendrites and dendritic spines) was interpreted as evidence for an important function of these receptors in modulating the excitatory glutamatergic input to the striatum [24]. 

D_2_ receptor immunoreactivity was detected by immunocytochemistry and electron micrography in rat basal ganglia. Subcellular experiments using fusion protein antibodies depicted predominant localization of D_2_ in spiny dendrites and spine heads within the neutrophil of the striatum. The receptors were also located in submembranous sites of dendritic shafts and dendritic spines [83].

## 4. A_2A_R and D_2_R Co-Aggregate, Co-Internalize and Co-Desensitize

Interactions between A_2A_ and D_2_ receptors were found to affect not only the signaling but also the intracellular trafficking of the two proteins. The human neuroblastoma cell line SH-SY5Y constitutively expresses A_2A_ receptors. In a groundbreaking article [84], SH-SY5Y cells were transfected with D_2_ receptors, and incubated with fluorescein-conjugated anti-A_2A_R (green fluorescence) and rhodamine-conjugated anti-D_2_R antibodies (red fluorescence). Receptor trafficking in the cells could then be monitored with confocal laser microscopy. In untreated cells, A_2A_R and D_2_R were shown to be generally at close distance (<100 nm) but rather uniformly distributed in the plasma membrane. When the cells were treated with either CGS21680 (100 nM) or quinpirole (10 µM) for 3 h, the distribution of the receptors in the plasma membrane became less uniform and significant co-aggregates were formed (yellow hotspots). When the same doses of the A_2A_R and D_2_R agonist were administered together for 3 h, the total intensity of the fluorescence signals was decreased, suggesting that co-aggregation of the A_2A_R and D_2_R was followed by co-internalization. This effect was dose-dependent, both the co-aggregation and the signal loss being stronger after treatment with 200 nM CGS21680 plus 50 µM quinpirole than with 100 nM CGS21680 plus 10 µM quinpirole. In cells lacking D_2_R, quinpirole did not cause any aggregation or internalization of the A_2A_R. Prolonged (3 h) administration of either 1 µM of CGS21680 or 1 µM of quinpirole to cells expressing both A_2A_R and D_2_R resulted in desensitization of their A_2A_ receptors (decrease of the cAMP response to A_2A_R stimulation), but desensitization of the D_2_R occurred only when both agonists were simultaneously administered [84]. When A_2A_R in the cells were immunoprecipitated with A_2A_R antibodies, Western blots indicated that the D_2_R was co-precipitated and that three glycosylated forms of the D_2_R were present in the precipitate [84]. Thus, A_2A_R and D_2_R were shown to co-aggregate, co-internalize, co-desensitize, and co-precipitate in the presence of D_2_R and A_2A_R agonists. 

Computer-assisted analysis of dual-channel fluorescence laser microscopy images indicated co-localization, co-aggregation and co-internalization of A_2A_R and D_2_R also in Chinese hamster ovary (CHO) cells [85,86]. In the CHO cell experiments, the effect of receptor stimulation was examined at different time intervals (3, 15 and 24 h) after administration of quinpirole. Co-aggregation of A_2A_R and D_2_R was observed after 3 h, and the co-aggregates internalized after 15 h. A return to the plasma membrane was detected after 24 h. In contrast to treatment with quinpirole, treatment of CHO cells with the D_2_R antagonist raclopride did not decrease but increased the fluorescence signal of both A_2A_R and D_2_R, indicating that a D_2_R antagonist reduced the internalization of the two receptors [86].

Similar microscopy techniques suggested that A_2A_ and D_2_ receptors form a macrocomplex with caveolin-1 that internalizes when cells are treated with an A_2A_ and a D_2_ agonist. Thus, caveolin-1 may play a role in the process of co-internalization [87]. Later experiments using bioluminescence resonance energy transfer (BRET) indicated that A_2A_ and D_2_ receptors also form a macrocomplex with ß-arrestin2, A_2A_R agonists promoting (and A_2A_R antagonists reducing) the D_2_R agonist-induced recruitment of ß-arrestin2 by the D_2_R protomer and subsequent co-internalization [88,89]. 

The D_2_R agonist 3-(3,4-dimethylphenyl)-1-(2-piperidin-1-yl)ethyl)-piperidine was shown to reduce the affinity and functional responsiveness of A_2A_R to agonists. In addition, this D_2_R agonist induced co-internalization of the A_2A_R and D_2_R proteins [90].

## 5. A_2A_R and D_2_R Are at Very Close Distance in Biomembranes and Form Heteromers

At the end of the twentieth and beginning of the twenty-first century, several biophysical techniques, such as atomic force microscopy (AFM), bimolecular fluorescence complementation (BiFC), fluorescence resonance energy transfer (FRET), bioluminescence resonance energy transfer (BRET), in situ proximity ligation assay (PLA), and AlphaScreen technology, were developed that allow the detection of spatial proximity of protein molecules, and such techniques have also been applied to A_2A_ and D_2_ receptors [85,91,92,93,94,95,96,97,98,99,100]. The results of these techniques and the observed co-aggregation, co-internalization and co-immuno-precipitation of A_2A_R and D_2_R indicate that both receptors are at very close distance in biological membranes (<10 nm) and form heteromers. Molecular biology experiments have provided insight in the mechanisms and atomic interactions that are involved in heteromer formation.

Using BRET technology, Japanese authors demonstrated that A_2A_R form homomers and also heteromers with D_2_R in living HEK293T cells. A_2A_ and D_2_ receptors were fused to either an energy donor (Renilla luciferase) or an energy acceptor (modified green fluorescent protein) without affecting the ligand binding affinity, subcellular distribution or co-immunoprecipitation of the two receptor proteins [101]. BRET and FRET techniques were also applied to quantify A_2A_R/D_2_R heteromers in receptor co-transfected cells, including cells that were transfected with modified D_2_ receptors: Chimeric proteins in which part of the D_2_ receptor protein was replaced by the corresponding part of the D_1_ receptor protein. Such experiments, and molecular modeling studies, suggested that heteromerization between A_2A_R and D_2_R depends on interaction of the third intracellular loop of the D_2_R with the C-terminal tail of the A_2A_R [102,103]. Transmembrane domains of the D_2_R, particularly the fifth transmembrane domain, also appeared to play a role [37]. A comprehensive molecular model of the A_2A_R/D_2_R heteromer was developed [104].

Triplet homologies in A_2A_R and D_2_R (e.g., alanine-alanine-arginine) have been proposed to guide the heteromer partners and to clasp them together [105]. “Pull-down” assays are in vitro methods to identify and determine physical interactions between two proteins. Using such techniques and mass spectrometry, a strong electrostatic interaction was demonstrated between negatively charged motifs (aspartic/phosphorylated serine residues) in the C-terminal tail of the A_2A_R and a positively charged (arginine-rich) epitope in the third intracellular loop of the D_2_R [106]. This electrostatic interaction was shown to possess an amazing stability, comparable to the stability of a covalent bond [107]. The importance of the serine residue in the C-terminal tail of the A_2A_R for A_2A_R-D_2_R receptor–receptor interaction was proven by mutation studies. A point mutation (change of serine 374 to alanine) reduced the formation of A_2A_/D_2_ heteromers and the allosteric modulation of D_2_R by A_2A_R agonists and antagonists [108]. Additional mutation of two aspartate residues (401–402 to alanine) in the C-terminal tail of the A_2A_R reduced the heteromer formation even further and completely abolished the allosteric modulation of D_2_R by A_2A_ ligands [109]. The importance of transmembrane domains of the D_2_R for heteromer formation was proven by administering synthetic peptides corresponding to the structure of the fourth and fifth transmembrane domain of the D_2_R. Such peptides reduced the ability of A_2A_R and D_2_R to form heteromers [109]. BRET techniques also demonstrated that calmodulin (CaM) interacts with the C-terminal tail of the A_2A_R and provided evidence for the formation of CaM-A_2A_R-D_2_R oligomeric complexes [110].

Japanese investigators created a single-polypeptide chain A_2A_R/D_2_R heteromer by fusing the C-terminus of the A_2A_R to the N-terminus of the D_2_R via a type II transmembrane protein. The resulting synthetic heterodimer showed similar specific binding of A_2A_R and D_2_R ligands and functional coupling to G-proteins as the original wild-type receptors [111].

A very interesting study used BiFC to demonstrate the presence of receptor oligomers in CAD cells, a differentiated neuronal cell model. Prolonged treatment of the cells with the D_2_R agonist quinpirole led to internalization of D_2_R/D_2_R oligomers and A_2A_R/D_2_R heteromers and decreased the relative number of A_2A_R/D_2_R heteromers compared to A_2A_R/A_2A_R oligomers. This effect of quinpirole was reversed by D_2_R antagonists (spiperone, sulpiride), and prolonged treatment of the cells with either a D_2_R antagonist or the A_2A_R agonist MECA resulted in a significant increase of the relative number of A_2A_R/D_2_R heteromers compared to A_2A_R/A_2A_R oligomers. Changes of the heteromer:oligomer ratio were not equivalent to the changes of total A_2A_R and D_2_R numbers in the cells. Thus, drug treatment appeared to modulate G-protein-coupled receptor oligomerization [112].

Investigators from Taiwan demonstrated that both A_2A_R and D_2_R are substrates for sialyltransferases (e.g., St8sia3) in the mouse striatum. If sialylation is reduced (as in St8sia3 knockout mice), a larger fraction of both receptors moves to lipid rafts and a greater number of D_2_R form heteromers with A_2A_R. Thus, sialylation may be a mechanism counteracting heteromer formation and shifting the homomer/heteromer equilibrium in the living brain [113]. Treatment of mice with an A_2A_R antagonist (SCH58261) causes a dose-dependent increase of locomotor activity. This response is much lower in St8sia3-knockout animals than in wild-type mice [113]. On the other hand, treatment of mice with a D_2_R antagonist (L741626) results in a dose-dependent reduction of their locomotor activity, and St8sia3-knockout animals are more sensitive to this effect of a D_2_R antagonist than their wild-type counterparts [113]. Alterations of the A_2A_R/D_2_R homomer/heteromer equilibrium in the striatum thus appear to be associated with altered responses of the animals to adenosine and dopamine receptor blockade. 

D_2_R-agonists can inhibit the 4-aminopyridine-provoked glutamate release in rat striatal astrocytes. Modulation of this inhibition with CGS21680 was shown to depend on the formation of A_2A_R/D_2_R heteromers, whereas the synthetic peptide VLRRRRKRVN abolished the effect of CGS21680 [81,114]. VLRRRRKRVN binds to the region of the D_2_R that is involved in electrostatic interaction with the A_2A_R and thus blocks the formation of A_2A_R-D_2_R heteromers [106]. 

Using fluorescent PLA and time-resolved FRET, A_2A_R/D_2_R complexes were detected in the striatum of rodents [94,115,116,117], monkeys [118], and humans [119]. Such complexes could also be demonstrated and quantified in postmortem brain tissue from patients with Parkinson’s disease and healthy control subjects, using AlphaScreen technology [97].

A_2A_R/D_2_R heteromers are now considered to be receptor heterotetramers, consisting of an A_2A_R homodimer and a D_2_R homodimer, each coupled to its own G-protein (Gs and Gi, respectively). Adenylate cyclase subtype AC5 also forms part of this multi-protein complex [41,120,121,122,123,124,125]. The heterotetramer model can explain the apparently paradoxical effects of A_2A_R agonists and antagonists on D_2_R ligand binding in CHO cells that were described in Section 2.3 of this review. Occupancy of the A_2A_R homodimer by either an agonist or an antagonist (at high dose) causes a conformational change in the heterotetramer, resulting in decreased function of the D_2_R protomer in the complex. However, when one of the two adenosine binding sites in the A_2A_R homodimer is occupied by an agonist and the other is simultaneously occupied by an antagonist, the conformational change does not occur [41].

## 6. Pharmacological Consequences of A_2A_/D_2_ Heteromer Formation

Receptors can form heteromers if certain basic criteria are met. These include: (a) The individual receptors that can form a heteromeric complex (protomers) must co-localize (i.e., be present in the same membrane domains, at very close distance from each other) and physically interact; (b) formed receptor complexes must exhibit distinct properties which differ from those of the individual, isolated protomers; and (c) chemical compounds that bind selectively to the heteromers should alter the properties or functions of the heteromers [126].

A_2A_R and D_2_R meet all these criteria. Biophysical and molecular biology techniques have demonstrated that these receptors co-localize and physically interact, both in cells and in mammalian tissues (see above, Section 5). Synthetic peptides that interact with the receptor domains involved in heteromer formation affect the electrostatic interactions between the protomers and alter the response of cells to certain drugs (Section 5). In addition, within A_2A_R/D_2_R heterotetramers, various receptor–receptor interactions are possible [125]:

(i) “Canonical interaction”. The agonist-activated Gi-coupled receptor in the complex (i.e., the D_2_R) will inhibit the activation of adenylate cyclase AC5 by the Gs-coupled receptor (i.e., the A_2A_R) [36]. The Ras GTPase domain of the subunits of the Gs and Gi proteins will interact with the C2 and C1 catalytic domains of adenylate cyclase AC5. The receptor partners in the complex can modulate each other’s downstream signaling cascade [36].

(ii) “Allosteric interaction”. Allostery is defined as communication between distant sites in a protein (or protein complex) in which energy associated with ligand binding or conformational change at one site is transferred to other, remote sites of the protein (or protein complex) resulting in changes of the kinetic or conformational properties of these sites. When a ligand binds to one of the receptors in an A_2A_R-D_2_R complex, the conformation of the complex (quaternary structure of the heterotetramer) is altered, resulting in different binding and signaling properties of the other receptor proteins in the complex [41,121,127,128,129]. When an A_2A_R ligand (either an agonist or an antagonist) binds to the A_2A_R homodimer in the complex, the affinity and signaling efficacy of D_2_R agonists is decreased. On the other hand, when a D_2_R agonist binds to the D_2_R heteromer in the complex, the binding of A_2A_R agonists is suppressed. Such allosteric effects between A_2A_R and D_2_R have been demonstrated in isolated biomembranes, intact cells, brain slices, and living animals (see above, Section 2).

(iii) “Formation of new modulatory sites”. When different receptor proteins associate to form a heteromer, novel binding sites may be created that are not present in the isolated receptors. Ligands specific to the receptor complex as such may exist [130] and, if discovered, may be used to specifically modulate the complex when it is present [131] (see also Section 11.3). 

(iv) “Higher order interaction”. A_2A_R/D_2_R heterotetramers may become part of higher order heteromers, so-called “receptor mosaics” [132]. Such interactions may, for example, involve the metabotropic glutamate receptor 5 (mGluR5) [133,134] or the sigma-1 receptor [117,135,136,137]. The presence or absence of such additional partners in a higher-order heteromer changes the strength of A_2A_R-D_2_R allosteric interactions and alters the response of the A_2A_ and D_2_ protomers to adenosine or dopamine. Since an unknown (and variable) number of additional proteins may bind to A_2A_R and D_2_R, the term “heteroreceptor complexes” is used in recent literature rather than A_2A_/D_2_R heterotetramers [138].

Thus, A_2A_R/D_2_R heterotetramers have a distinct pharmacology and distinct functions which differ from those of the individual constituent receptors [139].

## 7. A_2A_/D_2_ Interactions and Parkinson’s Disease

Upper motor neurons in the motor regions of the cortex initiate movements, such as continuous postural control, body locomotion, orientation towards sensory stimuli, and orofacial behavior. The activity of lower motor neurons in the spinal cord is coordinated by the upper motor neurons. These lower motor neurons directly or indirectly innervate skeletal muscle fibers [140].

In movement control, there is also a close cooperation of regions in the cortex with the basal ganglia [65,141] (Figure 2). Neurons that belong to the basal ganglia regulate the activity of the upper motor neurons although they do not directly project to them. The major nuclei that comprise the basal ganglia are: The striatum, the globus pallidus (GP), the substantia nigra (SN), and the subthalamic nucleus (STN) [142] (Figure 2). In the rodent brain, the striatum is a single nucleus whereas in primates, it is divided into caudate nucleus and putamen [143]. The basal ganglia receive input from areas of the cerebral cortex and their output is directed towards the thalamus, from where there is a transient excitation back to the motor regions in the cortex (Figure 2). MSNs in the striatum are known to be involved in movement control.

Activation of GABAergic MSNs of the “direct pathway” results in inhibition of the globus pallidus internus (GPi), for GABA is an inhibitory neurotransmitter. Since the GPi is connected to the thalamus via another GABAergic projection, inhibition of the GPi causes disinhibition of the thalamus. Because the thalamus contains excitatory neurons that project to the cortex, activation of the direct pathway results in facilitation of motor activity [140] (Figure 2).

In the “indirect pathway”, GABAergic MSNs project from the striatum to the globus pallidus externus (GPe). A second GABAergic projection runs from the GPe to the subthalamic nucleus (STN) and an excitatory glutamatergic projection connects the STN to the GPi. Activation of the indirect pathway therefore results in disinhibition of the STN and activation of the GPi. This activation of the GPi causes inhibition of the thalamus and reduced activity of the excitatory neurons that run from the thalamus to the cortex. Thus, activation of the indirect pathway results in suppression of motor activity. Although this description of the indirect pathway is probably a gross over-simplification [144], the concept is still widely used as a basis for research and therapy.

Normal movements require a delicate, coordinated balance of activity in the direct and indirect pathways [65]. The healthy brain contains dopaminergic neurons in the substantia nigra pars compacta that project to the striatum. Dopamine from these neurons stimulates the MSNs from the direct pathway via D1R and inhibits the MSNs from the indirect pathway via D2R. Both actions of dopamine facilitate motor activity. Loss of dopaminergic neurons from the brain, as occurs in Parkinson’s disease, will result in a decreased activity of the direct pathway, an increased activity of the indirect pathway and impaired motor control, particularly hypokinesia.

Since loss of dopamine results in overactivity of the indirect pathway, A_2A_R antagonists have been proposed as therapeutic drugs for the treatment of Parkinson’s disease [75,145,146,147,148,149]. Such drugs may restore the disturbed balance between the indirect and direct pathways and may increase the effect of endogenous dopamine, L-DOPA and specific D_2_ agonists, at least in the early stages of Parkinson’s disease [150,151]. A_2A_R antagonists may bind to the A_2A_R protomer in A_2A_R-D_2_R heteromeric complexes and increase the affinity of the D_2_R protomer for dopamine, its coupling to the G-protein and its signaling. In accordance with this hypothesis, perfusion measurements with MRI and pulsed arterial spin labeling have proven that the A_2A_R antagonist tozadenant inhibits (i.e., suppresses the overactivity of) the indirect pathway in the brain of Parkinson’s patients [152].

A_2A_R antagonists have been shown to be beneficial in various animal models of PD (e.g., D2R knockout mice [153], 6-OHDA-lesioned rats [154,155] and mice [156], rats with pharmacological D_2_R blockade [157], MPTP-treated marmosets [158], and MPTP-treated monkeys [159]). Since locomotor abnormalities in D_2_R knockout mice were rescued by the blockade of A_2A_R, not all actions of A_2A_R are related to the formation of A_2A_R-D_2_R heteromers. Apparently, striatal neuronal activity can also be regulated by A_2A_R via a dopamine D_2_R-independent pathway [152]. 

Many clinical studies have been performed to explore the effect of adenosine antagonists in Parkinson patients. These studies involved the non-subtype selective adenosine antagonists theophylline [160,161,162] and caffeine [163], and the A_2A_R-antagonists istradefylline [164,165,166,167,168,169,170,171,172,173,174] and tozadenant [175]. In a single study, theophylline was reported to have no significant effect, probably because group sizes were too small to reach adequate statistical power [162], but in two other studies, the drug caused mild improvement of the objective and subjective symptoms of disability and did not worsen dyskinesia [160,161]. Caffeine temporarily improved freezing of gait in Parkinson’s patients with symptoms of total immobility, but not in subjects who suffered from episodes of trembling with incapacity to any further movement [163]. Istradefylline as monotherapy was reported to not improve motor symptoms in early PD [169], but as adjunct therapy was shown to potentiate and prolong the action of L-DOPA. In the presence of istradefylline, lower doses of L-DOPA could be given to the patients and the severity of dyskinesia and resting tremor were reduced [164]. Several studies reported a reduction in “off” time (i.e., the time intervals in which disease symptoms return) when patients were given istradefylline [165,166,167,168,170,171] or tozadenant [175] in combination with L-DOPA, and this beneficial effect was not associated with any increase of dyskinesia [168]. Other symptoms of Parkinson’s disease, such as daytime sleepiness [172], gait disturbance, freezing of gait, and postural instability [174], were also improved by istradefylline. As a consequence of these positive findings, istradefylline is now a registered drug for treatment of Parkinson’s disease, both in Japan [176] and in the U.S. [177].

## 8. A_2A_R-D_2_R Interactions and Schizophrenia

Schizophrenia is thought to be associated with an overactivity of dopamine neurons in the ventral tegmental area of the brain, resulting in increased D_2_R signaling in the nucleus accumbens [178]. As explained above (Section 2.1 and Section 3.1), A_2A_R and D_2_R are present not only in the dorsal striatum, but also in the nucleus accumbens. Powerful antagonistic interactions between both receptors occur in this area of the brain and could be detected both in receptor binding studies and in microdialysis experiments. Administration of CGS21680 resulted in a reduced efficacy of dopamine to displace [^125^I]iodosulpiride from D_2_R in the nucleus accumbens. Infusion of the A_2A_R agonist CGS21680 in the nucleus accumbens had the same effect as infusion of the D_2_R antagonist raclopride (i.e., increasing the extracellular levels of GABA in the ipsilateral ventral globus pallidus), and the stimulation of GABA release by an A_2A_R agonist and a D_2_R antagonist were found to be synergistic [179]. 

According to several hypotheses, altered levels of extracellular adenosine and adenosine receptors are involved in the pathophysiology of schizophrenia [180,181,182]. In accordance with such hypotheses, A_2A_R were found to be upregulated in the striatum [183,184] and hippocampus [185] of chronic schizophrenics (although this upregulation could also be a consequence of the antipsychotic treatment that the patients received). A Chinese study reported significant associations between single nucleotide polymorphisms of the A_2A_R gene and schizophrenia in the northern Chinese Han population [186].

Since D_2_R of the ventral striatopallidal neurons are implied in the antipsychotic effects of neuroleptics [187], A_2A_R agonists, either alone or in combination with D_2_R antagonists, have been proposed as potential anti-schizophrenic drugs [179]. The ventral striatopallidal GABA pathway is considered as an anti-reward pathway which is over-activated in schizophrenia due to increased activation of its D_2_R [188]. The antagonistic A_2A_R-D_2_R interactions in the nucleus accumbens, which presumably occur within receptor heteromers, could be exploited to reduce the activity of the D_2_R protomer in the heteroreceptor complex [151,189]. In support of this idea, CGS21680 was shown to act as an atypical antipsychotic drug in rodent models of schizophrenia (phencyclidine, amphetamine) [190] and also in monkeys [191].

Some findings in humans have suggested that stimulation of A_2A_R may be beneficial in the treatment of psychosis. Dipyridamole, a nucleoside transport inhibitor that increases the extracellular levels of adenosine, has been tested as an add-on therapy in the treatment of schizophrenics. Combined treatment with haloperidol and dipyridamole (16 patients) was found to be significantly better than treatment with haloperidol and placebo (14 patients) in reducing positive and general psychopathology symptoms as well as PANNS scores [192]. Administration of allopurinol, a drug which blocks the degradation of purines and increases the levels of adenosine and inosine in the brain, resulted in clinical improvement in two poorly responsive schizophrenic patients [193].

Chronic treatment of rodents with clozapine, an atypical antipsychotic which is more effective than classical antipsychotics in some patients, was found to increase the activity of the enzyme ecto-5′-nucleotidase in the striatum, whereas chronic treatment with haloperidol did not have this effect [194]. These preclinical data suggest that clozapine treatment, in contrast to treatment with typical antipsychotics, is associated with increases of the levels of extracellular adenosine in the brain and with stimulation of A_2A_R.

## 9. A_2A_R-D_2_R Interactions and Treatment of Drug Addiction

According to a common hypothesis of reward-related behavior, the nucleus accumbens exerts tonic inhibitory effects on downstream structures in the brain. When MSNs in the nucleus accumbens are inhibited (e.g., by stimulation of dopamine D_2_R), these downstream structures are excited and an endogenous brake on reward-related behavior is released [195]. Addictive drugs are believed to be rewarding and reinforcing due to their effects on the dopamine reward pathway. They enhance dopamine release as is, for example, the case with nicotine, or they inhibit the reuptake of dopamine as does cocaine, or they act themselves as agonists at D_2_R [196].

Physiologically-relevant rewarding stimuli cause a release of dopamine in the shell of the nucleus accumbens, and this response is subject to habituation when the stimuli are repeatedly administered. Thus, the amount of dopamine that is released by a rewarding non-drug stimulus decreases as a result of repeated exposure to that stimulus. However, the dopamine response in the nucleus accumbens to addictive drugs is not prone to habituation but rather to sensitization, meaning that the amount of dopamine that is released by the drug increases as a result of repeated drug exposure.

Animal experiments in which rats with electrodes implanted in the medial forebrain bundle were trained to rotate a wheel in order to receive a rewarding electrical current have indicated that A_2A_R agonists elevate current reward thresholds (i.e., inhibit central reward processes) [197]. This observation suggests that A_2A_R modulate reward.

Studies in animal models of cocaine addiction have indicated that stimulation or blockade of A_2A_R has a significant impact on cocaine use. Administration of the non-subtype selective adenosine receptor antagonist caffeine to rats facilitates cocaine self-administration [198,199], whereas an A_2A_R agonist, like CGS21680 or NECA, suppresses the tendency of animals to take cocaine [200,201]. Adenosine receptor agonists appear to suppress cocaine intake by an interaction with A2AR in the nucleus accumbens, since microinjections of CGS21680 in the nucleus accumbens, but not in the prefrontal cortex, dose-dependently decrease cocaine self-administration [202]. Microinjections of a synthetic TM5 peptide (which interacts with the fifth transmembrane domain of the A_2A_R and disrupts A_2A_/D_2_ heterotetramers), completely counteracted the inhibitory effect of CGS21680 on cocaine intake [203]. In contrast to this striking impact of a TM5 peptide, microinjections of a TM2 peptide (which disrupts A_2A_/A_2A_ homodimers but not A_2A_/D_2_ heterotetramers) did not counteract the effect of CGS21680 on cocaine self-administration [204]. These results suggest that the beneficial actions of CGS21680 in animal models of cocaine abuse are mediated by the triggering of an allosteric inhibition of D_2_ protomer signaling in A_2A_R-D_2_R heteromeric complexes.

The development of cocaine sensitization is enhanced when rats are treated with the A_2A_R antagonist MSX-3 but is reduced when they are treated with the A_2A_R agonist CGS21680 or the D2R antagonist raclopride [205]. Administration of CGS21680 (0.25 to 0.5 mg/kg) to rats decreases the acquisition and expression of conditioned place preference induced by cocaine [206] or amphetamines [207]. 

In the treatment of substance abuse, relapse or drug-seeking behavior after a period of abstinence is a very serious problem. Thus, the finding that CGS21680 dose-dependently inhibits cocaine-induced reinstatement in rats after a period of drug abstinence of at least one week [201,208] is of great interest. On the other hand, A_2A_R antagonists (MSX-3, istradefylline, SCH58261, CGS15943), when administered systemically or by microinjections in the nucleus accumbens, promote cocaine-seeking behavior [202,209,210,211]. The impact of A_2A_R antagonists appears to be dependent on the question whether postsynaptic or presynaptic A_2A_R are blocked. Istradefylline is a postsynaptic A_2A_R antagonist, whereas SCH442416 blocks mainly presynaptic A_2A_R [212]. Postsynaptic blocking was found to enhance whereas presynaptic blocking reduced reinstatement of cocaine seeking [211,213]. The different antagonist affinities of pre- and postsynaptic A_2A_R may be due to the fact that presynaptic A_2A_R form heteromers with adenosine A_1_R, whereas postsynaptic A_2A_R interact with dopamine D_2_R.

Prolonged cocaine self-administration in rats is associated with a significant upregulation of A_2A_R in the nucleus accumbens [214,215]. After seven days of cocaine withdrawal, A_2A_R numbers in this area of the brain return to normal. This upregulation has been interpreted as a compensatory mechanism to counteract cocaine-induced increases in D_2_R signaling [214]. Mice that were prenatally exposed to cocaine showed an upregulation of D_2_R function and a downregulation of adenosine transporter function, consistent with increased levels of extracellular adenosine and more stimulation of A_2A_R [216]. Thus, cocaine exposure both prenatally and in later life, has direct effects on the dopamine and modulatory adenosine systems.

Cocaine is known to also increase the density of sigma-1R in the nucleus accumbens [217] and to cause trafficking of intracellular sigma-1R to the plasma membrane, where they can interact with D_2_R [135,218]. In fact, cocaine self-administration has been reported to increase the number of A_2_R-D_2_R and D_2_R-sigma-1R heteromers in the nucleus accumbens shell [117]. These data can also be interpreted as the formation of A_2A_R-D_2_R-sigma-1R heteromeric complexes in response to cocaine, the addition of the sigma-1R to the complex resulting in increased strength of antagonistic A_2A_R-D_2_R interactions [136,137,219,220].

BRET experiments in HEK-293T cells that were co-transfected with A_2A_R and D_2_R demonstrated that cocaine induces a concentration-dependent transient decrease of D_2_R homodimers and A_2A_R/D_2_R heteromers, but not of A_2A_R homodimers, via a specific interaction with the D_2_R. In co-transfected CHO cells, cocaine was found to cause an increase of the affinity of D_2_R for dopamine and increased coupling of D_2_R to G-proteins by changing the conformation of the receptor protein [221].

Based on such findings (and many others, which are extensively reviewed in [196]), it has been postulated that stimulation of A_2A_R could be a possible strategy to treat drug addiction [201,222,223,224]. A_2A_R antagonists that preferentially block presynaptic A_2A_R may also offer therapeutic benefits.

## 10. A_2A_R-D_2_R Interactions and Attention Deficit Hyperactivity Disorder

Attention-deficit hyperactivity disorder (ADHD) is a disorder of human behavior that involves dysfunctions of sustained attention, behavioral hyperactivity and impulsivity. ADHD seems to be characterized by reduced functioning of the dopaminergic reward pathway [225,226]. Oral methylphenidate, an inhibitor of noradrenaline and dopamine reuptake, is often prescribed as a therapeutic drug to treat ADHD.

A study that was published in 2000 reported that apart from several genes of the noradrenergic system, polymorphisms of the A_2A_R gene are significantly associated with human ADHD [227]. A later Swedish study confirmed that the A_2A_R gene may indeed be involved in ADHD traits [228]. In rodent models of ADHD, A_2A_R were found to be upregulated in various brain regions [229,230] and adenosine A_2A_R antagonists were shown to have beneficial effects, such as improvement of short-term object-recognition ability, attention and memory function [230,231] and improved development of frontal cortical neurons [232].

A large study involving 1239 human subjects reported an association between the rs2298383 TT genotype of the A_2A_R and anxiety disorders in ADHD. No association with the D_2_R genotype was detected, but a significant, positive gene-gene interaction effect between A_2A_R and D_2_R on the presence of anxiety disorders was noted [233]. This synergistic effect between the A_2A_R and D_2_R genes suggests that A_2A_R-D_2_R heteromers could be explored as a possible target in the treatment of ADHD.

## 11. PET Imaging of Adenosine–Dopamine Interactions

Positron emission tomography (PET) is a minimally invasive imaging technique that allows quantitative assessment of the interaction of radioactive ligands with receptors, enzymes, or transporters in the living brain. Since PET makes it possible to study such interactions repeatedly in experimental animals and humans, this imaging modality may be employed to acquire information about adenosine–dopamine interactions in the healthy human brain, their alterations in disease, and the impact of treatment. Radioligands for adenosine A_2A_ and dopamine D_2_ receptors are currently available (see Table 1 and Table 2, and [234,235,236,237,238] for an overview). However, until now the number of PET studies aiming to demonstrate A_2A_/D_2_ interactions have been very limited.

Based on findings acquired with other techniques and reported in the literature, three classes of PET studies concerning adenosine–dopamine interactions appear possible:

### 11.1. Pharmacological Challenge Studies

In these studies, subjects are scanned twice with a radioligand for adenosine A_2A_R or dopamine D_2_R, first at baseline (or after administration of a placebo) and then at follow-up, after a pharmacological challenge with a drug that binds to the other receptor system (a dopaminergic drug in the case of A_2A_R imaging, and a purinergic drug in the case of D_2_R imaging). Three investigations that used PET imaging have shown that this experimental set-up allows the detection of adenosine–dopamine interactions in the brain of living mammals.

In the first study [242], the radiotracer [^18^F]MRS5425, an analogue of the A_2A_R antagonist SCH442416, was used to image A_2_AR in the brain of rats that had been unilaterally lesioned with 6-hydroxydopamine. In this animal model of Parkinson’s disease, the authors observed an increased binding of the tracer in the ipsilateral (lesioned) striatum with respect to the contralateral (healthy) striatum. The increase of [^18^F]MRS5425 in the lesioned hemisphere suggests that loss of dopaminergic neurons can cause upregulation of postsynaptic D_2_ and A_2A_ receptors, and binding of the PET ligand [^18^F]MRS5425 may be used as a biomarker to monitor Parkinson’s disease. Some animals were subsequently treated with the dopamine D_2_R agonist, quinpirole. A significant (15–20%) decrease of the striatal uptake of [^18^F]MRS5425 was observed after acute administration of quinpirole. The decreased binding of the A_2A_R ligand after a dopaminergic challenge indicates that interactions between D_2_R and A_2A_R can be monitored in living animals with PET [242].

In the second study [381], healthy human subjects with low levels of daily caffeine intake received oral caffeine (300 mg) and the impact of this challenge on the dopaminergic system was assessed by measuring changes of the binding of [^11^C]raclopride to D_2_R in the brain. A small but significant increase in the binding potential of [^11^C]raclopride was detected in the putamen and ventral striatum (5 to 6%), but not in the caudate nucleus. The rise in the ventral striatum was associated with an increase of alertness caused by caffeine [381]. In an earlier study, which involved administration of 200 mg of oral caffeine to eight human subjects with higher levels of daily caffeine intake, a trend towards increased [^11^C]raclopride binding in the ventral striatum was also noted, but this did not reach statistical significance [382].

In the third study (which was performed in our own institution), anesthetized healthy rats received either the A_2A_R agonist CGS21680 (1 mg/kg, i.p.), the A_2A_R antagonist istradefylline (1 mg/kg, i.p.) or vehicle (saline) and the impact of these challenges on the dopaminergic system was assessed by PET imaging, using full kinetic modeling of the cerebral uptake of the radioligand [^11^C]raclopride. Significant decreases of [^11^C]raclopride binding potential were detected, which were strong (>50%) after intraperitoneal administration of CGS21680 and moderate (30%) after administration of istradefylline [383].

However, these studies also highlighted the complexity of interactions in the living brain and difficulties in pinpointing the exact mechanisms underlying the observed changes. Altered binding potentials in PET imaging may indicate: (i) An altered size of the total receptor population (i.e., altered expression of the receptor gene). (ii) An altered affinity of existing receptors for the radioligand (which may be due to allosteric receptor–receptor interactions within heteromeric complexes). Both A_2A_R agonists (like CGS21680) and A_2A_R antagonists (like istradefylline) can allosterically decrease the affinity of the D_2_R protomer for agonists and antagonists [41,129]. (iii) Increases or decreases of the fraction of internalized receptors (since, in most cases only receptors on the cell surface will bind the radioligand). The adenosine A_2A_R agonist CGS21680 promotes the recruitment of ß-arrestin2 to the D_2_R protomers in an A_2A_/D_2_ heteromer complex and causes subsequent co-internalization of A_2A_ and D_2_ receptors [84,88], a process in which caveolin-1 is involved [87]. (iv) Increases or decreases of the extracellular concentration of the endogenous neurotransmitter or neuromodulator (which competes with the radioligand for binding to a limited number of receptor sites). Selective adenosine A_2A_R antagonists may increase the release of dopamine [384] and may also inhibit the enzyme monoamine oxidase B and thus raise the levels of extracellular dopamine [385]. The first mechanism (altered gene expression) is unlikely as an explanation for the observed changes of [^11^C]raclopride binding potential, since the PET studies employed an acute drug challenge and measured radioligand binding shortly after the challenge. The increased binding potential of [^11^C]raclopride that was noticed in the ventral striatum after administration of caffeine cannot reflect a decrease of extracellular dopamine, since increased alertness was noticed under these conditions. Increased alertness is normally related to augmented release of dopamine in the striatum, whereas reductions of extracellular dopamine are accompanied by increased tiredness and sleepiness [381]. Thus, the increase of [^11^C] raclopride binding after caffeine intake may reflect an altered affinity of D_2_R for the radioligand or a reduced internalization of D_2_R in the presence of caffeine. 

The PET studies mentioned above [242,381,382,383] indicate that adenosine–dopamine receptor interactions can be visualized and quantified in the brain of living mammals, but various mechanisms or a combination of mechanisms may be involved and may cause the observed changes.

Other PET studies have indicated that antagonistic effects between adenosine A_2A_ and dopamine D_2_ receptors at the MSNs of the striatum occur at physiological levels of receptor occupancy in the living brain. The D_2_R antagonist haloperidol is widely used as an antipsychotic, but can induce extrapyramidal symptoms (i.e., movement disorders, such as catalepsy (rigidity, muscle stiffness, fixed posture)). In non-human primates, the duration of the cataleptic posture induced by haloperidol (0.03 mg/kg, i.m.) was reduced when animals were treated with the A_2A_R antagonist ASP5854 (0.1 mg/kg, oral). A PET study with the A_2A_R ligand [^11^C]SCH442416 showed that the anti-cataleptic effect of ASP5854 (0.1 mg/kg, oral) was reached at an A_2A_R occupancy of 85% [284].

In Parkinson’s patients treated with dopaminergic medication (and rodents with 6-OHDA induced hemiparkinsonism), A_2A_R were found to be upregulated if dyskinesia was present, but not when dyskinesia was absent [56,262,278,285]. This finding indicates that adenosine–dopamine interactions are clinically relevant and A_2A_R antagonists may be applied as therapeutic drugs [386].

### 11.2. Studies with Bivalent Radioligands

As discussed above, PET imaging with a suitable radioligand for adenosine A_2A_ or dopamine D_2_ receptors may be used to gain information on adenosine–dopamine interactions. Changes of radioligand binding to one protomer after a pharmacological challenge to the other protomer can be monitored with PET. The magnitude of these changes may be proportional to the relative abundance of A_2A_/D_2_R heterotetramers and/or the strength of the A_2A_R–D_2_R interaction, which could be altered by disease or after a successful treatment. 

Another approach to visualize and quantify A_2A_R/D_2_R heterotetramers is the use of radiolabeled bivalent ligands for PET imaging, so-called “bivalent probes”. In early attempts to target receptor homodimers, the orthosteric sites of two homodimer partners were bridged by a ligand consisting of two identical pharmacophores connected by a short linker. A similar approach could be tried to target heteromers. The two receptor partners in a heteromer may be bridged by a ligand consisting of two different pharmacophores (appropriately designed for each individual heteromer partner) connected via a short spacer. In this way, the ligand can bind simultaneously to two GPCR receptors if these receptors are closely together (i.e., within a receptor heteromer). A successful bivalent ligand will bind more avidly (with 10–100 fold greater affinity) to the appropriate receptor heteromer than to the isolated receptor monomers or homodimers. 

Some experience with bivalent ligands has been acquired by pharmacochemists. In the past, virtually all therapeutic drugs were designed to target a single protein. The discovery of heteromeric receptors has led to a new interest in the development of mixed action drugs for combination therapies, or drugs which selectively bind to receptor heteromers [138,387]. A heterobivalent ligand combining D_2_R agonism with A_2A_R antagonism could be an effective antiparkinsonian drug and might also be radiolabeled for PET imaging of A_2A_R/D_2_R heteromers [388]. The potential of such mixed-actions drugs has been demonstrated in the opioid system, where successful bivalent analgesics combining µ-agonism with δ-antagonism have been developed [389,390].

The synthesis of A_2A_ antagonist-D_2_ agonist heterobivalent drugs was first reported in 2009 [391]. The spacer in these drugs was based on trifunctional amino acids that were combined with PEG-polyamide unit repeats. Various bivalent ligands were constituted by connecting the A_2A_R antagonist 8-(p-carboxymethyloxy)phenyl-1,3 dipropylxanthine (XCC) and the D_2_R agonist (+/-)-2-(N-phenethyl-N-propyl)amino-5-hydroxytetralin (PPHT-NH2) via a Lys-Lys-[PEG/polyamide]n-Lys-Glu (n = 0–7) linker (Figure 3). In competition radioligand binding experiments using striatal membranes, it was shown that these bivalent ligands could displace specific A_2A_R and D_2_R radioligand binding. The bivalent compounds displaced the specific monovalent ligands [^3^H]-ZM241385 and [^3^H]-YM09151-2 only when both A_2A_ and D_2_ receptors were expressed in cells. Such displacement could also be observed in striatal tissue, indicating the presence of A_2A_/D_2_ receptor heteromers. This suggests that heterobivalent ligands could potentially serve as PET probes for A_2A_/D_2_ receptor heteromers in native tissues and as pharmacological tools to investigate the properties of A_2A_R-D_2_R heterotetramers. They could also pave the way for the design of heteromer-selective drugs for the treatment of Parkinson’s disease.

The distance between the ligand pharmacophores should correspond to the distance of the two binding sites in the receptor heteromer. Thus, the pharmacophore units should be connected with a linker of the appropriate length. Docking experiments predicted that a linker of 26 atoms allows two pharmacophore moieties to bind to the A_2A_/D_2_ heteromer complex. The points of attachment of the linker to the pharmacophore units are another crucial aspect of bivalent ligand design. During the development of a bivalent ligand with A_2A_R antagonist and D_2_R agonist action, it was found that the [COOH] position of the adenosine antagonist XCC and the N-terminal position of dopamine D2R agonist were most suitable for the attachment of a linker [391]. Another difficulty in the design of bivalent ligands is the fact that the attachment of a linker and a spacer generally results in a reduction in affinity of each pharmacophore for its target. Even if the lead compounds of bivalent ligands have target affinities in the low nanomolar range, the fused ligand with its spacer may have a too low affinity for successful PET imaging [392]. Lead compounds with very high affinities may be required for the design of a successful bivalent probe for PET imaging.

Bivalent ligands also face challenges concerning CNS uptake, metabolism, and excretion. CNS drugs and radioligands for CNS imaging must cross the blood–brain barrier (BBB) in order to be effective. The ability of compounds to penetrate the BBB by passive diffusion is guided by a number of molecular properties: Topological polar surface area (tPSA) should be <90 Å, hydrogen bond donors (HBDs) < 3, cLog P = 2–5, and the molecular weight <450 Da [393]. Although some properties of bivalent and heterobivalent drugs may not satisfy these criteria, they may still cross the BBB. The serotonin receptor agonist sumatriptan has a molecular weight of 721 Da (i.e., greater than 450 Da), yet it crosses the BBB and is a successful therapeutic drug [392]. Thus, although the design of a bivalent A_2A_R/D_2_R ligand remains a serious challenge, the problems may be overcome.

Another approach to development of bivalent A_2A_R-D_2_R ligands has been described in the literature [394]. This approach resulted in a new compound, DP-L-A2AANT, that was prepared by amide conjugation of dopamine (DP) to an A_2A_ antagonist (A2AANT) via a succinic spacer (L) (Figure 4). The spacer was bound to the amine group of A2AANT. The fusion compound showed a high A_2A_R affinity (K_i_ 2.07 ± 0.23 nM in rat striatum). Although the drug did not exhibit a high affinity towards D_2_R and cannot be considered as a suitable candidate for labeling with a positron emitter for PET imaging, it could be a lead compound for the development of antiparkinsonian drugs and PET tracers. Administration of DP-L-A2AANT led to a release of L-A2AANT and dopamine that could be detected in heparinized human whole blood after one and two hours, and DP-L could be detected after 8 h. The bivalent drug may allow prolonged delivery of small amounts of dopamine which are associated with little neuronal toxicity and limited side effects compared to conventional dopaminergic treatments, especially since succinic acid is known to have low toxicity in humans [394].

The potent A_2A_R antagonist ZM 241385 and the D_2_R agonist ropinirole are considered to have favorable properties for the design of bivalent therapeutic drugs. These classic ligands were used in 2015 to synthesize a novel series of compounds [395] (Figure 5). A cyclic linker between the A_2A_ and D_2_ pharmacophores could be used to increase the structural rigidity of a bivalent ligand. Generally, triazine-linker based members of the family showed a 4-fold decrease in A_2A_ inhibitory potency compared to the parent A_2A_R antagonist and maintained their functional potency towards the D_2_R. Non-cyclic dual acting ligands showed a 28- to 54-fold reduction in their A_2A_R inhibitory potency. Many of the developed compounds passed preliminary BBB permeability tests. However, the in vivo brain uptake kinetics of these dual acting ligands should still be determined [395].

In conclusion, although the development of bivalent radioligands for PET imaging of A_2A_R/D_2_R heterotetramers will be a major challenge, the design of such compounds may prove to be possible.

### 11.3. Studies with Radiolabeled Heteromer-Specific Allosteric Modulators

Experiments in which the impact of homocysteine on A_2A_R-D_2_R heteromers was examined have suggested that heteromer-specific allosteric modulators may exist [131,220,396,397]. In CHO cells that express both A_2A_R and D_2_R, homocysteine reduces the internalization of A_2A_R-D_2_R complexes after stimulation of the D_2_R [396]. Homocysteine was shown to form a non-covalent complex with an arginine-rich epitope involved in heteromer formation but did not disrupt or prevent the formation of A_2A_R-D_2_R heteromers in co-transfected HEK cells [396]. In striatal astrocytes, homocysteine reduces the D_2_R-mediated inhibition of glutamate release but does not affect the A_2A_R-mediated antagonism of this D_2_R effect [397]. These data have been interpreted as evidence that homocysteine binds to A_2A_R-D_2_R heteromers and modulates the allosteric energy transmission between A_2A_R and D_2_R in a heteromer complex [220].

Labeling homocysteine with a positron emitter is definitely not a viable strategy to develop a heteromer-specific radioligand for PET imaging, but if other substances can be identified which bind to specific pockets in the A_2A_R–D_2_R interface within a heteromer without disrupting or preventing heteromer formation, such substances could be used as lead compounds to develop heteromer-specific imaging agents or therapeutic drugs.

### 11.4. Other Opportunities for PET Imaging

PET may not only be used to visualize and quantify A_2A_R-D_2_R heterotetramers or to determine the strength of receptor–receptor interactions within heteromeric complexes, but also to assess the physiological consequences of pharmacological targeting of A_2A_R-D_2_R heteromers. Regional changes of cerebral glucose metabolism after administration of a heterobivalent drug may be measured with the PET tracer [^18^F]fluorodeoxyglucose (FDG), and regional changes of cerebral perfusion can be measured with flow tracers and PET or single photon emission computed tomography (SPECT), or with functional magnetic resonance imaging (fMRI). In animal models of Parkinson’s disease, A_2A_R antagonists have been found to not only improve motor function, but also to be neuroprotective [398,399,400,401,402,403]. The neuroprotective actions of such drugs, drug combinations or heterobivalent drugs may be assessed with PET imaging (e.g., by visualizing dopaminergic nerve endings with dopamine transporter ligands such as [^11^C]PE2I or [^18^F]-FE-PE2I, or by visualizing neuroinflammation with radiolabeled TSPO ligands, in longitudinal studies. An exciting final possibility is to compare changes of the binding of a radioligand for presynaptic A_2A_ receptors, such as [^11^C]SCH442416, with changes of the binding of a radioligand for postsynaptic A_2A_ receptors in disease models (although [^11^C]KW6002 = istradefylline is probably not ideal for this purpose).

## 12. Conclusions

Understanding the complexities of A_2A_/D_2_ interactions is important in unravelling basal ganglia physiology. Initial observation of antagonistic interactions between adenosine and dopamine in membrane preparations, intact cells and living animals was followed by proof of direct interactions between A_2A_ and D_2_ receptors supplied by various biophysical techniques and demonstration of the molecular mechanisms involved in these protein-protein interactions. Such biochemical and biophysical findings facilitated further studies of basal ganglia disorders.

Several neurodegenerative and neuropsychiatric disorders are associated with a dysregulation of corticostriatal and nigrostriatal afferents that leads to aberrant neurotransmission. Membrane interactions between adenosine A_2A_R and dopamine D_2_R are an important aspect of striatal function and appear to be altered in Parkinson’s disease, schizophrenia, substance abuse, and ADHD. A_2A_R and D_2_R have been proposed as therapeutic targets. Decreasing A_2A_ signaling by selective A_2A_ antagonists may result in a recovery of GPe activity in Parkinson’s disease, thereby reinstating the thalamocortical motor stimulatory activity. A_2A_ agonists, either alone or in combination with D_2_R antagonists, have been proposed for the treatment of schizophrenia. Such combination treatments reduce overactivity of the D_2_R protomer in the A_2A_/D_2_ receptor complex. Drugs that are known to increase the levels of extracellular adenosine (such as nucleoside transport inhibitors, inhibitors of purine degradation and antipsychotics increasing the activity of the enzyme 5′-nucleotidase) may be used as a replacement for A_2A_ agonists in A_2A_R-D_2_R based combination therapies. Stimulation of A_2A_R has also been postulated as a possible strategy to treat substance abuse, in particular addiction to cocaine. A_2A_R antagonists, on the other hand, could have a beneficial effect in combination therapies for ADHD. Thus, heteromers of A_2A_R and D_2_R are potential targets for the treatment of several human disorders.

PET imaging may provide significant in vivo information that leads to greater understanding of the role of A_2A_R/D_2_R heteromers in the physiology of the healthy and diseased brain. PET studies with radioligands for A_2A_R or D_2_R before and after a pharmacological challenge to the other protomer in the A_2A_R-D_2_R complex may be used to assess the strength of A_2A_/D_2_ receptor interactions. Bivalent ligands that bind simultaneously to A_2A_ and D_2_ receptors if the receptor proteins are at close proximity may be used as molecular probes to assess the regional abundance of A_2A_R-D_2_R heteromers. The physiological consequences of pharmacological targeting of A_2A_R-D_2_R heteromers in patients can be assessed by PET imaging with tracers visualizing cerebral energy metabolism, cerebral perfusion, neuroinflammation, or dopamine transporter expression.

## Figures and Tables

**Figure 1 ijms-22-01719-f001:**
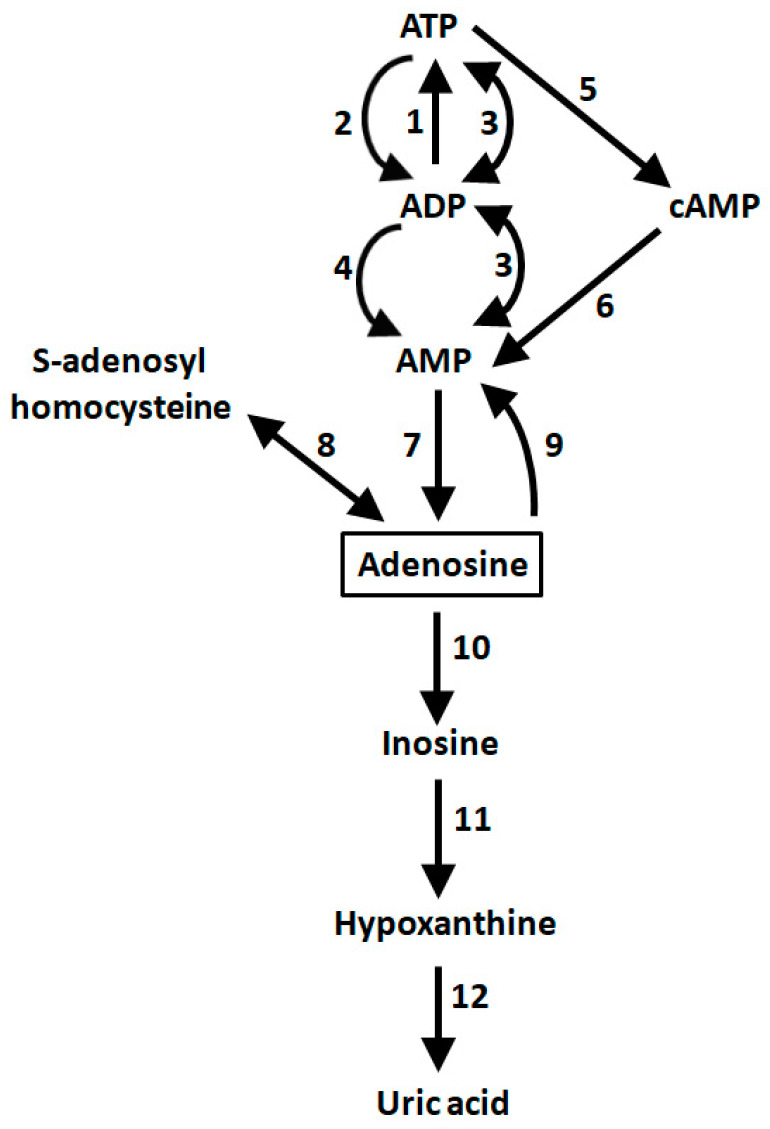
Metabolic pathways involved in the formation and removal of adenosine. 1 = Oxidative phosphorylation (and creatine kinase), 2 = Energy-consuming processes, 3 = Adenylate kinase, 4 = Apyrase, 5 = Adenylate cyclase, 6 = Phosphodiesterase, 7 = 5′-Nucleotidase, 8 = S-adenosyl homocysteine hydrolase, 9 = Adenosine kinase, 10 = Adenosine deaminase, 11 = Purine phosphorylase, 12 = Xanthine oxidase. ATP = adenosine 5’-triphosphate, ADP = adenosine 5’-diphosphate, AMP = adenosine 5’-monophosphate, cAMP = 3’.5’-cyclic adenosine monophosphate.

**Figure 2 ijms-22-01719-f002:**
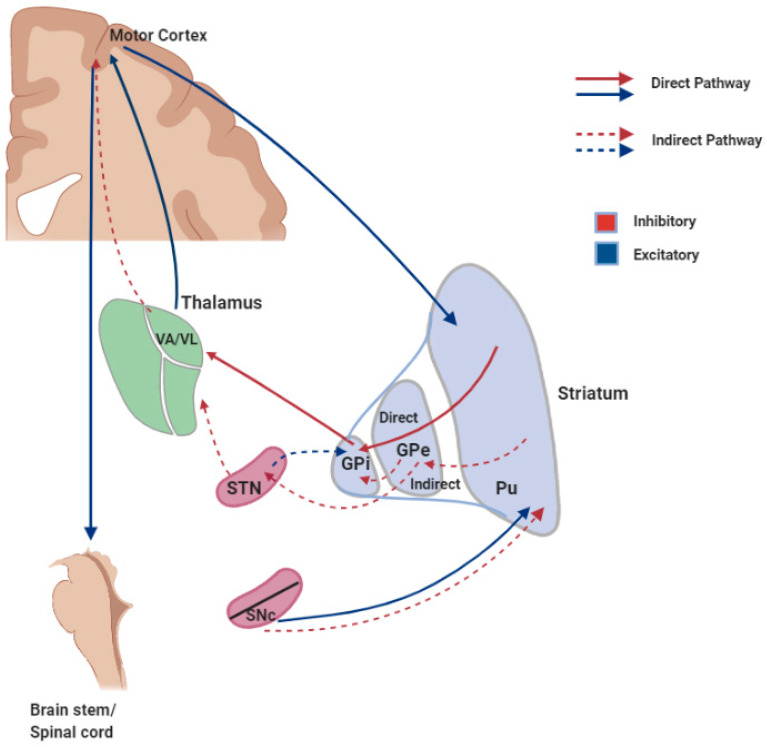
Schematic drawing of the direct and indirect pathways for motor control. Solid and faded lines represent direct and indirect pathways, respectively. Blue lines represent excitatory connections and red lines represent inhibitory connections. Pu = putamen, Gpe = globus pallidus externus, Gpi = globus pallidus internus, STN = subthalamic nucleus, SNc = substantia nigra pars compacta, VA/VL = ventral anterior/ventral lateral thalamic nucleus. Created with BioRender.com.

**Figure 3 ijms-22-01719-f003:**
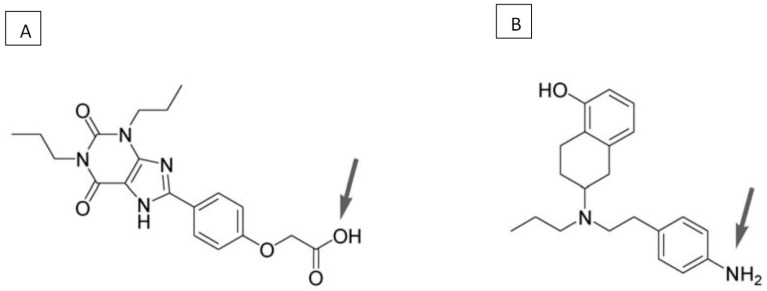
Chemical structures of the A_2A_ antagonist XCC (**A**) and the D_2_ agonist PPHT-NH2 (**B**). By attaching a linker to the atomic positions indicated by the arrowheads, a bivalent ligand for A_2A_R/D_2_R heteromers can be created [392].

**Figure 4 ijms-22-01719-f004:**
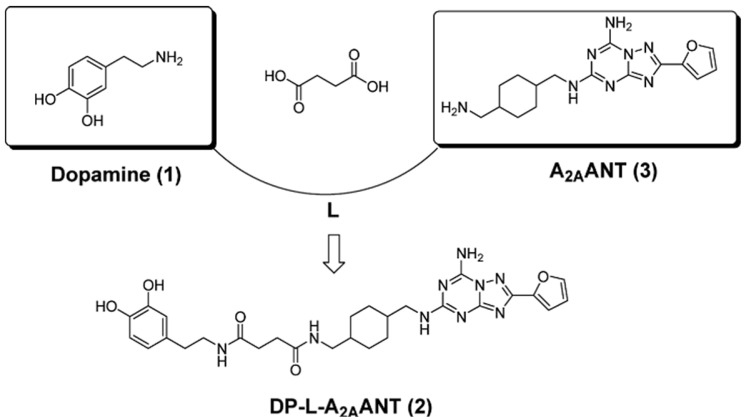
Conjugation of dopamine with the A2A antagonist 7-amino-5-(aminomethyl)-cyclohexylmethyl-amino-2-(2-furyl)-1,2,4-triazolo[1,5-a]-1,3,5-triazine trifluoroacetate via a succinate spacer, to obtain the prodrug DP-L-A2AANT, a bivalent ligand [394].

**Figure 5 ijms-22-01719-f005:**
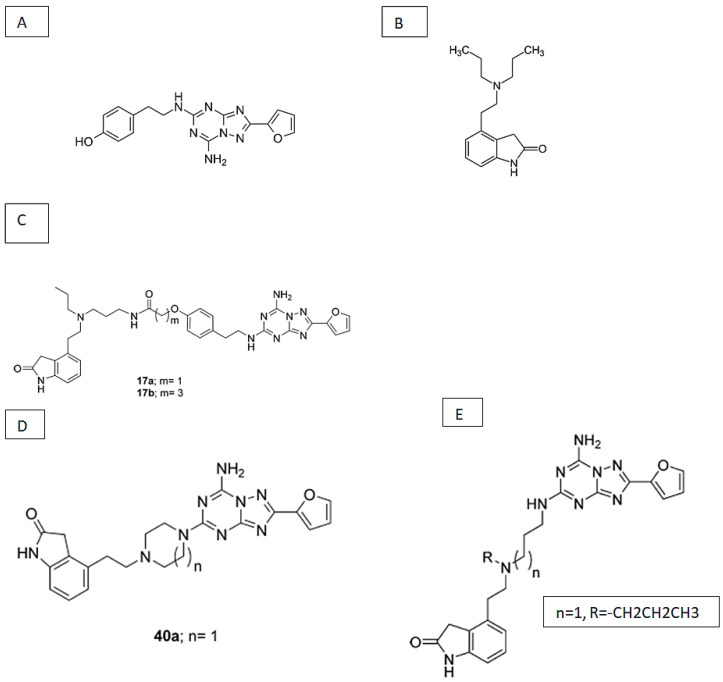
Conjugation of the adenosine A_2A_R antagonist ZM241385 (**A**) and the dopamine D_2_ agonist ropinirole (**B**) to form a heterobivalent ligand (**C**). Dual action drugs can be prepared by using cyclic (**D**) or non-cyclic (**E**) spacers, and the latter may contain an ionizable tertiary amine [395].

**Table 1 ijms-22-01719-t001:** Overview of ligands for positron emission tomography (PET) imaging of A_2A_ receptors.

Ligand (Alphabetic Order)	Animal Study		Human Study	Comments
	Animal Model	Reference		
[^11^C]CSC	Rodent	[239]		
DMPX analogues	Rodent	[240]		
[^18^F]FDA-PP_1_ [^18^F]FDA-PP_2_				Agonists, no in vivo data [241]
[^18^F]FESCH (=[^18^F]MRS5425)	Rodent	[242,243,244,245]		
[^11^C]Istradefylline (=[^11^C]KW6002)	Rodent	[246,247]	[247]	Extrastriatal off-target binding
[^11^C]KF17837	Rodent	[248,249,250]		High non-specific binding
	Monkey	[251]		
[^11^C]TMSX (=[^11^C]KF18446)	Rodent	[252,253,254,255,256,257]	[257,258,259,260,261,262,263,264,265,266,267,268,269,270]	
[^11^C]KF21213	Rodent	[271]		
[^18^F]MNI-444	Monkey	[272,273]	[274]	
[^11^C]Preladenant	Rodent Monkey	[275,276,277,278] [279]	[280,281]	
[^11^C]SCH442416	Rodent Monkey	[282,283] [282,284]	[285,286]	

CSC = 8-(3-Chlorostyryl)caffeine, DMPX = 3,7-Dimethyl-1-propargylxanthine, TMSX = [7-methyl-11C]-(E)-8-(3,4,5- trimethoxystyryl)-1,3,7-trimethylxanthine. Other compounds are numbered by the producing institutions or pharmaceutical companies.

**Table 2 ijms-22-01719-t002:** Overview of ligands for positron emission tomography (PET) imaging of dopamine D_2/3_ receptors.

Ligand (Alpha- Betic Order)	Rodent, Pig or Cat Study	Monkey or Baboon Study	Human Study	Comments
[^18^F]Benperidol		[287,288]		
[^18^F]DMFP	[289,290,291]		[292,293]	Longer half-life than [^11^C]raclopride
N-Ethyl-[^11^C]- eticlopride		[294]		
[^11^C]Fallypride		[295]	[296]	
[^18^F]Fallypride	[291,297,298,299]	[297,300,301,302,303,304]	[305,306,307,308,309,310,311]	High-affinity, visualizes also extrastriatal D_2_R, numerous studies *
[^18^F]FCP		[312]		
[^18^F]FEBF	[313]			
[^18^F]FESP	[314,315]	[314,315,316]	[314,317]	
[^11^C]FLB457	[318]	[319,320]	[296,321,322,323,324,325,326,327,328]	High-affinity, visualizes also extrastriatal D_2_R, numerous studies *
[^11^C]FLB524		[329]	[329]	
5-[^18^F]FPE		[330]		
[^18^F]FPSP	[315,331]	[315,331]	[331]	
[^18^F]Haloperidol	[332,333]	[334]		Binds also to sigmaR
[^18^F]MABN	[335]	[336]		
[^18^F]MBP	[335]	[312,336]		Binds also to rho1
Methyl-[^11^C]- eticlopride		[294]		
[^11^C]MNPA	[337]	[338,339]	[340]	Agonist ligand
[^11^C]Nemonapride	[341,342,343]			Binds also to sigmaR
[^11^C]NMSP	[341,344,345]		[346,347,348,349]	
[^18^F]NMSP	[350]	[336,350]	[351]	
[^11^C]NPA	[352,353]	[352,354,355]	[356,357,358]	Agonist ligand
[^11^C]PPHT	[359,360]	[359]		Agonist ligand
[^11^C]Raclopride	[361,362]	[363,364,365,366,367]	[368,369,370,371,372,373,374,375,376,377,378,379]	Moderate affinity, visualizes mainly striatal D_2_R, numerous studies *
[^18^F]Spiperone		[287]		
[^11^C]SV-III-130		[380]		Partial agonist ligand
[^11^C]ZYY339	[359,360]	[359]		Agonist ligand

* Only a small selection of the available publications is cited for this radioligand. DMFP = Desmethoxyfallypride, FCP = Fluoroclebopride, FEBF = Fluorethyl-2,3-dihydrobenzofuran, FESP = Fluoroethyl- spiperone, FPE = Fluoropropyl-epidepride, FPSP = Fluoropropyl-spiperone, MABN = 2,3-dimethoxy-N-[9-(4-fluorobenzyl) -9-azabicyclo[3.3.1]nonan-3beta-yl]benzamide, MBP = 2,3-dimethoxy-N-[1-(4-fluorobenzyl)piperidin4yl]benzamide, MNPA = Methoxy-N-n-propylnorapomorphine, NMSP = N-methyl-spiperone, NPA = N-n-propylnorapomorphine, PPHT = (+/−)-2-(N- phenethyl-N-propyl)amino-5-hydroxytetralin. Other compounds are numbere by the producing institutions or pharmaceutical companies.

## Data Availability

Not applicable.

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
