# Peer review of "Allosteric Interactions between Adenosine A2A and Dopamine D2 Receptors in Heteromeric Complexes: Biochemical and Pharmacological Characteristics, and Opportunities for PET Imaging"

_ijms, 2021, doi:10.3390/ijms22041719_

Round 1

Reviewer 1 Report

The present review is interesting, well written, and provides an extensive body of information about the interactions between A2A adenosine receptors and D2 dopamine receptors from the discovery to the present. Furthermore, in the last part of the manuscript, the authors propose some interesting approaches for the design of novel imaging agents. I believe this manuscript is worthy of publication in the International Journal of Molecular Science but, minor revision is required.

In my opinion, table 1 could be more understandable to the readers if the columns dedicated to animal models were merged into one, and the animal model used was specified each time with its reference next to it.

Secondly, given the complexity and the richness of information in the manuscript, it would be appropriate to improve the conclusion section, reporting some more details of what has been discussed throughout the text, to make the point for readers and give them a more complete take-home message.

Written English is good but a general review is recommended.

Author Response

RESPONSE TO REVIEWER 1

"In my opinion, table 1 could be more understandable to the readers if the columns dedicated to animal models were merged into one, and the animal model used was specified each time with its reference next to it". --> We have re-organized Table 1 as reviewer 1 suggested.

"Secondly, given the complexity and the richness of information in the manuscript, it would be appropriate to improve the conclusion section, reporting some more details of what has been discussed throughout the text, to make the point for readers and give them a more complete take-home message". --> We have greatly expanded the conclusion section, and have reported more details, as reviewer 1 demanded.

"Written English is good but a general review is recommended". --> We have amended several sentences in the manuscript, in response to the comments of reviewer 2.

Reviewer 2 Report

The authors have undertaken a fairly comprehensive review of allosteric interactions between adenosine A2A and dopamine D2 receptors in brain.  The authors introduced antagonistic interactions between adenosine and dopamine.  Then, they overviewed distribution, interaction, the molecular mechanisms, and pharmacological consequences of A2A and D2 heteromer formation.  In addition, they discussed that the interactions are useful to be exploited in novel strategies for the treatment of Parkinson’s disease, schizophrenia, substance abuse, and attention-deficit/hyperactivity disorder.  Furthermore, they presented PET imaging studies of adenosine-dopamine interactions.  The authors concluded that A2A and D2 heteromer formation is a potential target for the treatment of disorders and PET imaging may provide significant information in the physiology of the health and disease in brain.  The manuscript is well-written.  I have some comments that I believe need to be addressed prior to publication of this article.

Comments:

Some sentences should be revised.

Page 1 lines 10–11, “This interaction involves (among others) adenosine A2A and dopamine D2 receptors (R).”

Page 1 lines 11–13, “Stimulation (or blockade) of A2AR inhibits (or enhances) D2R-mediated locomotor activation and goal-directed behavior in rodents.”

Page 1 lines 14–17, “Reciprocal A2AR-D2R interactions occur mainly in striatopallidal GABAergic medium spiny neurons (MSNs) of the indirect pathway involved in motor control, in MSNs of the nucleus accumbens involved in reward-related behavior, and in striatal astrocytes.”

Page 4 lines 146–147, “Yet, CGS21680 and caffeine canceled out each other’s effect on D2R affinity when they were administered together! [41].”

Page 6 lines 184–185, “Faded lines represent direct pathway. Solid lines represent indirect pathway.”, Solid and faded lines represent direct and indirect pathways, respectively.

Page 7 line 242, “< 100 nM”, < 100 nm

Author Response

RESPONSE TO REVIEWER 2

"Some sentences should be revised.

Page 1 lines 10–11, “This interaction involves (among others) adenosine A2A and dopamine D2 receptors (R).” --> We changed this sentence to:  "These interactions are mediated via adenosine A2A and dopamine D2 receptors (R)".

Page 1 lines 11–13, “Stimulation (or blockade) of A2AR inhibits (or enhances) D2R-mediated locomotor activation and goal-directed behavior in rodents.” --> We changed this sentence to: "Stimulation of A2AR inhibits and blockade of A2AR enhances D2R-mediated locomotor activation and goal-directed behavior in rodents".

Page 1 lines 14–17, “Reciprocal A2AR-D2R interactions occur mainly in striatopallidal GABAergic medium spiny neurons (MSNs) of the indirect pathway involved in motor control, in MSNs of the nucleus accumbens involved in reward-related behavior, and in striatal astrocytes.” --> We have split and rephrased the sentence as follows: "Reciprocal A2AR-D2R interactions occur mainly in striatopallidal GABAergic medium spiny neurons (MSNs) of the indirect pathway that are involved in motor control, and in striatal astrocytes. In the nucleus accumbens, they also take place in MSNs involved in reward-related behavior".

Page 4 lines 146–147, “Yet, CGS21680 and caffeine canceled out each other’s effect on D2R affinity when they were administered together! [41].” --> We removed the exclamation mark at the end of the sentence.

Page 6 lines 184–185, “Faded lines represent direct pathway. Solid lines represent indirect pathway.”, Solid and faded lines represent direct and indirect pathways, respectively. --> The reviewer is right, we have changed text to: "Solid and faded lines represent direct and indirect pathways, respectively".

Page 7 line 242, “< 100 nM”, < 100 nm --> The reviewer is right (nm means nanometer, whereas nM means nanomol). We have corrected this typing error.
